# Fatigue Reliability Assessment of an Automobile Coil Spring under Random Strain Loads Using Probabilistic Technique

**Reza Manouchehrynia, Shahrum Abdullah \* and Salvinder Singh Karam Singh**

Centre for Integrated Design for Advanced Mechanical System (PRISMA), Faculty of Engineering and Built Environment, Universiti Kebangsaan Malaysia, UKM Bangi, Selangor 43600, Malaysia; r.manouchehrynia@gmail.com (R.M.); salvinder@gmail.com (S.S.K.S.)
\* Correspondence: shahrum@ukm.edu.my; Tel.: +60-(3)-8911-8411

**Abstract:** This paper presents a mathematical model to estimate strain-life probabilistic modeling based on the fatigue reliability prediction of an automobile coil spring under random strain loads. The proposed technique was determined using a probabilistic method of the Gumbel distribution for strain-life models of automobile suspension systems. Strain signals from different road excitations in experimental tests were measured. The probability density function of the Gumbel distribution was considered to estimate model parameters using maximum likelihood estimation (MLE). The Akaike information criterion (AIC) method was performed to specify which model can estimate the best fit model parameters. Results demonstrated a good agreement between the predicted fatigue lives of the proposed probabilistic model and the measured strain fatigue life models. The root-mean-square errors (RMSE) based on the Coffin–Manson, Morrow, and Smith–Watson–Topper strain-life models were approximately 0.00114, 0.00107, and 0.00509, respectively, indicating a high correlation with the proposed model and experimental data. The results demonstrated that the proposed probabilistic model is effective for the fatigue life prediction of automobile coil springs using strain and stress fatigue life approaches.

**Keywords:** fatigue; maximum likelihood estimation; probabilistic modeling; strain load; reliability

## 1. Introduction

In recent years, random loadings have become important in predicting mechanical and structural system responses in various engineering fields. In this respect, the behavior of automobile suspension systems, specifically coil springs, can be influenced by different road excitation loadings. Automobile engineers perform probabilistic modeling for most vehicle parts owing to random road excitations, which can predict the reliability assessment of system failure. Probability and reliability analyses consider the quantitative and qualitative approaches of a system; however, the reliability assessment of fatigue characteristics of materials is important for production assessment [1].

Reliability assessment has been conducted using experimental, numerical, and analytical approaches alone and in combination [2]. This process generally includes fatigue life techniques, which have been evaluated by numerous researchers [3–5]. For instance, accelerated life is a technique to predict fatigue life and assess acceleration coefficients using a compilation of fatigue data and acceleration load spectra. Overs-peed testing is an innovation method to analyze stochastic finite elements for the prediction of quantifying uncertainties for material characterizations, loads, and experimental results. Zhu et al. [6,7] established a probabilistic framework based on a numerical approach to measure material and load variations for reliability assessment and fatigue life prediction.

These researchers combined mean stress and load variation effects on the basis of the strength of damage interference by using finite element simulations to evaluate the influence of fatigue life reliability. Azrulhisham et al. [8] described an approach to evaluate the fatigue life reliability of a steering knuckle subjected to road surface vibration loadings by using repeated load data based on a durability rig test for a passenger automobile. Kang et al. [9] estimated the fatigue life reliability of a steel structure based on a probabilistic distribution to predict residual fatigue life and the number of cycles to failure by using a fatigue stress-life model. The reliability of automobile suspension systems has advanced in terms of design optimization techniques based on low-cycle strain–fatigue life models [10]. Similarly, Song et al. [11] proposed a design optimization model for a vehicle-steering knuckle undergoing loading conditions of bump and brake components by using the least squares method.

Strain or stress fatigue life models can be predicted by probabilistic methods for fatigue failure criteria in new fatigue analysis approaches [12,13]. Given that the fatigue results are related to the deterministic data of an experimental test, the fatigue results can be investigated with probability and statistical methodologies. Moreover, most fatigue results indicate non-Gaussian (non-normal) and nonstationary distributions, particularly for vibration fatigue analysis [14] in studies on automotive fields. Nieslony et al. [15] assessed fatigue life for non-Gaussian loading signals using a combination of the spectral method and the Dirlik method. In this study, the Dirlik and spectral methods were used to evaluate the experimental results obtained with the rainflow cycle technique. Results showed that non-Gaussian loadings can be assessed using the spectral method in the absence of mean stress effect. Cianetti et al. [16] represented a new correction coefficient to predict fatigue life for a mechanical component based on non-Gaussian stress in the frequency domain. The proposed procedure illustrated that the correction coefficient can be used to estimate fatigue damage on the basis of stress-time histories affected when the kurtosis value is low. While, fatigue damage results were overestimated in the case of non-Gaussian stress because a new correction coefficient was required for implementation. The relationship of nonstationarities and non-Gaussianities was studied by Cesnik and Capponi in a vibration fatigue analysis. The Gaussianity and stationarity were important assumptions of the fatigue damage theory in the frequency domain approach [17,18]. Capponi indicated that different rates of amplitude-modulated nonstationary excitation have a shorter fatigue life than the stationary excitation level for the dynamic structure's response and dynamic loading.

In non-Gaussian distributions, the fatigue life reliability and probability of structures are usually analyzed using statistical strain- and stress-life models, such as the Gumbel, Weibull, Gamma, lognormal, and logistic distributions. Anderson [19] proposed a statistical method to represent stress–fatigue life based on the Gumbel distribution. This researcher determined the influence of scale and location parameters on fatigue life prediction. Results illustrated the effect of imperfections on initial crack lengths using the Gumbel distribution performance. In the case of strain or stress fatigue life prediction-based statistics, the existing model parameters must be estimated. In this regard, maximum likelihood estimation (MLE) is the most broadly used when material properties are desirable in test statistics [20,21]. In this case, the compatibility degree of stress range (simulation) and probability distribution (theory) parameters was determined to be approximately 0.834 for the Gumbel distribution [22]. Therefore, the advantages of the Gumbel model are as follows: (i) elastic–plastic local strain presents a mathematical (analytical) probabilistic description of an entire strain-life model, (ii) low- and high-cycle fatigue regions consider run out and failed fatigue life data, (iii) variables can be considered dimensionless to indicate a connection with any other initial variables, and (iv) the model can simplify damage analysis.

The present research proposes a mathematical model based on the strain-life probabilistic model of the reliability assessment of an automobile coil spring using the Gumbel distribution. Experimental strain-time history signals are measured from rural, campus, and highway road excitation loadings by using a data acquisition system. A novel methodology is generated to evaluate strain-life curves based on a probabilistic model in relation to the assumptions of the Gumbel model parameters for the Coffin–Manson, Morrow, and Smith–Watson–Topper (SWT) strain life models. Therefore, the

MLE method is used to estimate the parameters of the Gumbel distribution. Fatigue life prediction based on the reliability assessment of an automobile coil spring has been studied in recent years; however, the Gumbel distribution model has rarely been used.

## 2. Methodology

The probabilistic strain fatigue life prediction for an automobile coil spring is summarized in Figure 1. Three road excitations were measured from the vehicle coil spring. Then, the Gumbel model was compared with the Gaussian and Gamma models by using the Akaike information criterion (AIC) to obtain best fit model parameters. The Gumbel strain-based probabilistic fatigue model and generalized strain fatigue life models based on a probabilistic distribution were established to estimate probabilistic $\varepsilon$-N curves. The MLE method was performed to estimate scale and location parameters from the probability density function (PDF) of the Gumbel distribution to obtain probabilistic $\varepsilon$-N curves. Finally, fatigue life predictions were evaluated using the Pearson correction coefficient (PCC) and $R^2$ methods.

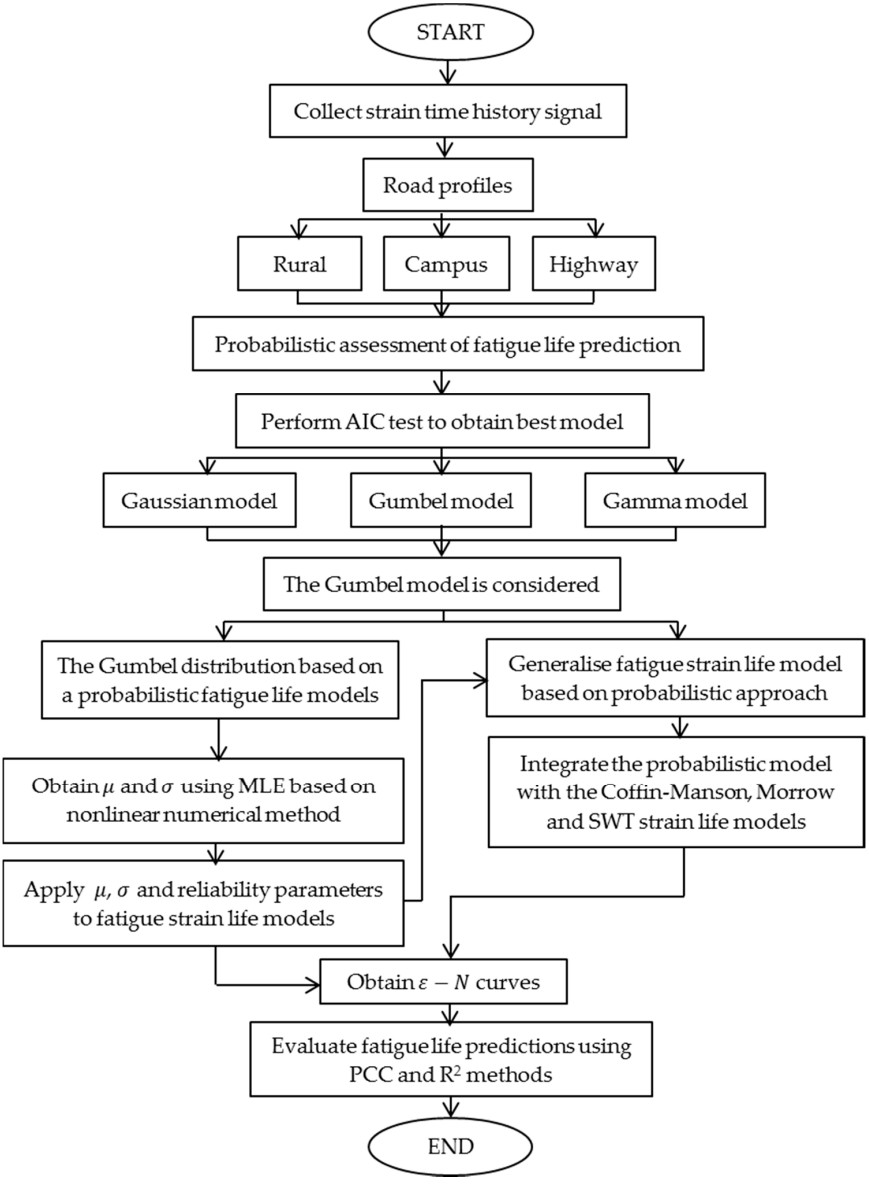

**Figure 1.** Framework of probabilistic model for strain fatigue life prediction ($\varepsilon$-N).

The automobile coil spring was selected to assess fatigue life reliability based on different road excitations for fatigue life prediction. Rural, campus, and highway road surfaces were considered to examine the responses of the coil spring when an automobile passed through various road profiles. Therefore, the three road excitations were investigated to ensure high amplitude activities of strain signals. Rural and campus road conditions represented high amplitude activities for a vehicle suspension system, whereas the highway road profile was considered as a smooth and well-maintained surface. The setup for measuring strain signals is shown in Figure 2. A data acquisition system was used to analyze the strain signals collected from a vehicle coil spring for rural, campus, and highway road excitations. A 500 Hz sample rate was selected for 100 s to obtain strain-time history signals. Thus, the strain signals were guaranteed appropriate for using a frequency rate of more than 400 Hz to prevent lost strain signals. From Figure 2, it can be seen that the coil spring surface was polished and scrubbed with a sand paper [23]. Therefore, the coil spring surface was smooth and there was no crack on the surface. A strain gauge was attached to the surface of the vehicle coil spring and then connected to a data logger system to analyze strain-time history signals for the three road profiles. The cyclic and fatigue properties of SAE 5160 alloy carbon steel were considered the mechanical properties of the coil spring, as shown in Table 1 [24].

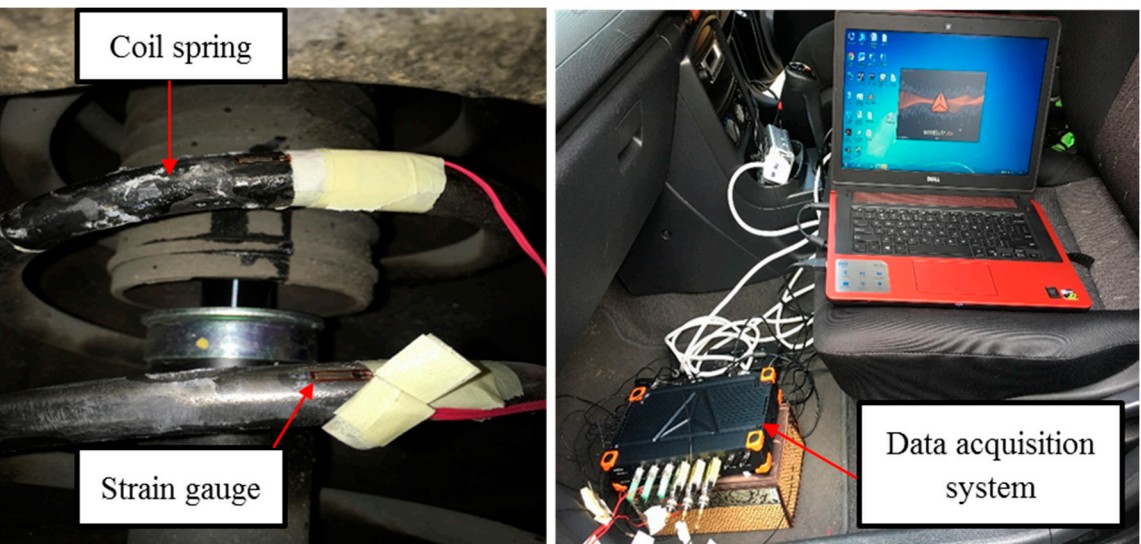

**Figure 2.** Experimental setup for measuring strain-time history signal.

**Table 1.** Material properties of SAE 5160 carbon steel.

| Properties | Values |
|---|---|
| Yield strength (MPa) | 1070 |
| Ultimate tensile strength (MPa) | 1550 |
| Material modulus of elasticity (GPa) | 207 |
| Fatigue strength coefficient (MPa) | 2063 |
| Fatigue strength exponent | −0.08 |
| Fatigue ductility exponent | −1.05 |
| Fatigue ductility coefficient | 9.56 |

In the case of fatigue life design, several stages were considered. The criteria of fatigue design were derived from infinite life to damage (defect) tolerance. Therefore, the criteria of fatigue design include the utilization of four fatigue life models: (i) stress-life (*S-N*) approach that cannot be used in durability analysis because this approach includes two parts—crack initiation and crack propagation, (ii) strain-life (*ε-N*) approach that has suitable usage in durability analysis, especially based on low-cycle fatigue because it considers only fatigue crack initiation (nucleation) and the durability assessment of automotive components are carried out based on safe life, (iii) fatigue crack growth (*da/dN − Δk*), and

(iv) a two-stage method using a combination of (ii) and (iii) [25–27]. Thus, this study took into account only fatigue crack initiation (nucleation) as fatigue failure is correlated to the localized plasticity in low-cycle fatigue. Figure 3 illustrates a schematic of failure analysis based on total life. Hence, the research scope has been determined by safe life region (before failure).

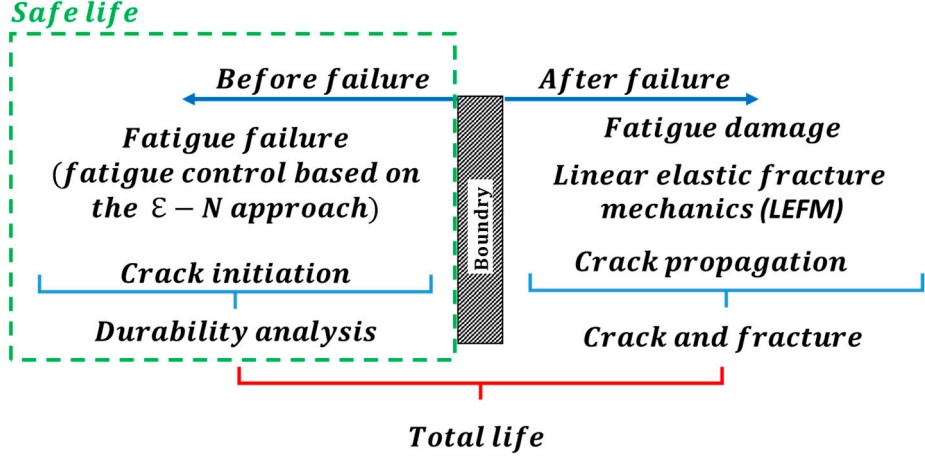

**Figure 3.** Schematic of fatigue failure analysis.

### 2.1. Gumbel Distribution Based on Probabilistic Assessment of Fatigue Life Model

A probabilistic approach can help understand reliability assessments. In this case, a probabilistic fatigue analysis was crucial to integrate several uncertainties arising from probabilistic methods, such as the material properties and geometric characterizations of components [28,29]. In this section, the Gumbel distribution based on a strain fatigue life model ($\varepsilon$-$N$) for probabilistic analysis is presented. This model is applicable for fatigue stress- and strain-life data with quantile curves without the need to disconnect from two parts of the total strain (elastic and plastic). In addition, the model determines physical and statistical assumptions to integrate elastic–plastic parts into a single model based on an analytical probabilistic description. The PDF for the Gumbel distribution is generally given by:

$$f(x,\, \sigma, \mu) = \frac{1}{\sigma} exp - \left(\frac{x-\mu}{\sigma}\right) exp\left[-exp - \left(\frac{x-\mu}{\sigma}\right)\right];\ x\epsilon\mathbb{R}\,,\quad \sigma > 0,\quad \mu\epsilon\mathbb{R} \tag{1}$$

where $x$, $\sigma$, and $\mu$ are independent variable, scale, and location parameters that assume that $f$ is the general formula for the PDF of the Gumbel distribution. Figure 4 shows the schematic of the PDF of the Gumbel distribution plot.

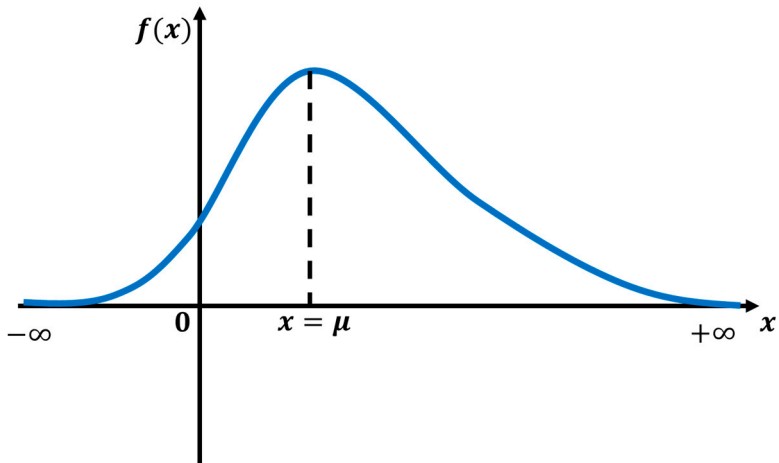

**Figure 4.** Schematic of the Gumbel probability density function distribution.

Accordingly, $N_f$ (number of cycles to failure) and $\varepsilon_a$ (total strain amplitude) were determined for strain amplitude and life with random variables. Therefore, the random variables can be written as $\widetilde{N}_f = \frac{N_f}{N_0}$ and $\widetilde{\varepsilon}_a = \frac{\varepsilon_a}{\varepsilon_0}$ to indicate dimensionless and specify the relationship between the initial variables. Several physical (e.g., limit range of variables and the weakest link principle) and statistical states (e.g., limit behavior and stability) should be considered to establish compatibility between strain or stress range and life [30]. In the strain-life field, the life of the cumulative distribution function (CDF) $H(\widetilde{N}_f; \widetilde{\varepsilon}_a)$ according to the strain range should be compatible with the strain range of the CDF given life $L(\widetilde{\varepsilon}_a; \widetilde{N}_f)$. In this regard, the functional equation can be written for the compatibility condition as $H(\widetilde{N}_f; \widetilde{\varepsilon}_a) = L(\widetilde{\varepsilon}_a; \widetilde{N}_f)$ [31]. This process can lead to the functional equation of the Gumbel strain-based fatigue model with the change of variable, and the new probability function is as follows:

$$P = H(\widetilde{N}_f; \widetilde{\varepsilon}_a) = \exp\left[ -\exp -\left\{ \frac{\log(N_f/N_0)\log(\varepsilon_a/\varepsilon_0) - \mu}{\sigma} \right\} \right], \tag{2}$$

where $P$ is the probability of failure based on the CDF of the Gumbel distribution, $N_0$ and $\varepsilon_0$ are threshold values of the life and endurance limit of $\varepsilon_a$, respectively; and $x$ expresses random sample of the model. Therefore, the random fatigue life ($N$) was substituted to the random sample of model ($x$) that is shown in the logarithmic form. Furthermore, Equation (2) indicates, in a dimensionless form, the probability of failure $P$ depends on the product of $\log\left(\frac{N_f}{N_0}\right)$ and $\log\left(\frac{\varepsilon_a}{\varepsilon_0}\right)$ only, that is, $\log\left(\frac{N_f}{N_0}\right)\log\left(\frac{\varepsilon_a}{\varepsilon_0}\right) \sim G(\mu, \sigma)$ using the Gumbel distribution. Therefore, all parameters ($N_0$, $\varepsilon_0$, $\sigma$, $\mu$) were easily obtained by using several established methods proposed in the fatigue literature [32].

## 2.2. MLE Method to Estimate the Gumbel Model Parameters

Several methods can be used to estimate the parameters of the proposed model. Nevertheless, the most popular method to estimate the suggested model parameters is the MLE method, which represents desirable statistical properties [31]. Consider $\varepsilon_i$ and $N_i$, ($i = 1, 2, \ldots, n$) as a set of random variables, where $\varepsilon_i$ and $N_i$ are the deterministic strain amplitude and random fatigue life based on PDF distribution [33]. Therefore, the Gumbel model parameters can be estimated via the MLE method from the log-likelihood function, which is given by:

$$L(\sigma, \mu) = -\sum_{i=1}^{n} \frac{N_i - \mu}{\sigma} - nLn(\sigma) - \sum_{i=1}^{n} exp -\left( \frac{N_i - \mu}{\sigma} \right), \tag{3}$$

The partial derivatives were considered as:

$$\frac{\partial LnL(\sigma,\mu)}{\partial \mu} = \frac{1}{\sigma}\left[ n - \sum_{i=1}^{n} exp -\left( \frac{N_i - \mu}{\sigma} \right) \right],$$

$$\frac{\partial LnL(\sigma,\mu)}{\partial \sigma} = \sum_{i=1}^{n}\left( \frac{N_i - \mu}{\sigma^2} \right) - \frac{n}{\sigma} - \sum_{i=1}^{n}\left( \frac{N_i - \mu}{\sigma^2} \right)exp -\left( \frac{N_i - \mu}{\sigma} \right), \tag{4}$$

by solving $\frac{\partial LnL(\sigma,\mu)}{\partial \mu} = \frac{\partial LnL(\sigma,\mu)}{\partial \sigma} = 0$ for $\sigma \neq 0$. The MLE method estimates $\sigma$ and $\mu$ using the iterative nonlinear numerical technique (Newton–Raphson method) of Equations (5) and (6).

$$\overline{N} - \frac{\sum_{i=1}^{n}(N_i)exp -\left( \frac{N_i}{\sigma} \right)}{\sum_{i=1}^{n} exp -\left( \frac{N_i}{\sigma} \right)} - \sigma = 0, \tag{5}$$

$$\mu - \sigma\left[ Ln(n) - Ln \sum_{i=1}^{n} exp -\left( \frac{N_i}{\sigma} \right) \right] = 0, \tag{6}$$

where $\overline{N}$ is the sample mean of the random fatigue life. The parameter of $\sigma$ was estimated explicitly using Equation (5), and $\mu$ was estimated in Equation (6) when the estimation of $\sigma$ was obtained [34,35].

### 2.3. Proposed Mathematical Model based on Probabilistic for Strain Fatigue Life Models

Most existing well-known strain fatigue life models are determined based on empirical methods and deterministic with the plot of the strain-life relationship. Although fatigue life data were obtained based on random data, the results were evaluated in a normal distribution by using regression and least square methods. Therefore, this normal distribution cannot be justified properly owing to the weakest link principle [32]. Probabilistic and statistics analyses are important methods to assess fatigue failures to prevent uncertainties which can occur in a component [36]. Therefore, the aim of this section was to establish a probabilistic approach on the basis of the Gumbel distribution model by using Coffin–Manson, Morrow, and SWT strain fatigue life models.

The unbiased estimator ($\hat{\mu}$) of a random variable was determined according to each sample equal to the estimated population parameter for the Gumbel distribution to estimate the point of the population parameter. In other words, the sample mean ($\overline{x}$) was considered as the unbiased estimator for the location parameter ($\mu$). Obtaining the value of the sample mean that is equal to the location parameter is important. Therefore, the sample mean can satisfy the location parameter estimator when the condition is unbiased ($\overline{x} = \hat{\mu}$). If fatigue life followed the Gumbel distribution ($N_1, N_2, \ldots N_i$), then $\mu$ of the Gumbel distribution was estimated via the sample mean of $\overline{N}$ as follows:

$$\hat{\mu} = \overline{N} = \frac{1}{n} \sum_{i=1}^{n} N_i \,, \tag{7}$$

where $n$ is the sample size of random variables. The sample scale parameter ($\sigma$) tends to find a population scale parameter of an unbiased estimator to delete bias by using a random sample distribution for fatigue life reliability design. Thus, the unbiased estimator of the Gumbel population scale parameter can be defined as follows:

$$\hat{\sigma} = \beta \sigma \,, \tag{8}$$

where $\hat{\sigma}$ and $\beta$ are the unbiased scale parameter estimator (or population) and the correction coefficient of the standard deviation.

Let $x_p$ be a random population of the interval estimation for the Gumbel distribution model, which is generally defined in probability as follows:

$$p\big(X > x_p\big) = \int_{x_p}^{\infty} f(x)dy = p \,, \tag{9}$$

where $f(x)$ is the PDF of random variable $X$. Therefore, the value of the population percentile (or true value) is defined by Equation (10), which is related to the reliability level of $p$ that is determined as safe life or safe fatigue strength in logarithmic form.

$$x_p = \mu + u_p \sigma \,, \tag{10}$$

where $\mu$ and $\sigma$ are location and scale parameters of the Gumbel distribution, respectively, and $u_p$ is the standard Gumbel distribution of the probability of $p$.

According to Figure 4, a percentile estimator, which is also referred to as a sample percentile, was considered and determined via Equation (11).

$$\hat{x}_p = \hat{\mu} + u_p \hat{\sigma} = \hat{\mu} + u_p \beta \sigma \,, \tag{11}$$

where $\hat{\mu}$ is an unbiased estimator of the location parameter of the Gumbel distribution. In this regard, the sample percentile cannot be equal with the population percentile amount. In other words, the sample percentile can be fewer or more than the true population. Consequently, an error level can consistently occur between the sample and the population percentiles. Figure 5 represents the probability density curve of fatigue life. The vertical axis and horizontal axis in Figure 5 represent the probability density of failure and fatigue life $N$, respectively.

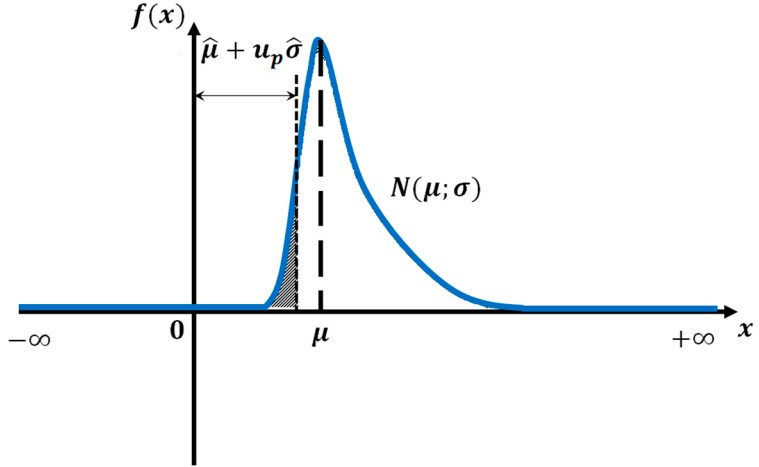

**Figure 5.** Probability density curve of fatigue life of interval estimation.

In Equation (8), the variation coefficient and the variance of the sample random variable were considered, and the $t$-distribution of the Gumbel model is written as follows:

$$t = \frac{\left(\hat{\mu} + u_p\hat{\sigma}\right) - \left(\mu + u_p\sigma\right)}{\sigma\sqrt{\frac{1}{n} + u_p^2(\beta^2 - 1)}} , \tag{12}$$

In the case of the confidence level of $\gamma$, the amount of $t_\gamma$ was calculated in an interval of $(-t_\gamma, t_\gamma)$, which is related to a confidence level of $\gamma$, as defined in Equation (13).

$$-t_\gamma < \frac{\left(\hat{\mu} + u_p\hat{\sigma}\right) - \left(\mu + u_p\sigma\right)}{\sigma\sqrt{\frac{1}{n} + u_p^2(\beta^2 - 1)}} < t_\gamma , \tag{13}$$

According to the $t$-distribution theorem of the Gumbel model, one side illustrated the lower confidence limit based on natural logarithmic fatigue life, which is related to a confidence level of $\gamma$, as follows:

$$p\{t < t_\gamma\} = \gamma , \tag{14}$$

where $t_\gamma$ is defined as the $\gamma$ percentile of $t$-Gumbel distribution representing a confidence level. When the population of the PDF curve demonstrates a non-normal distribution, estimating via a bootstrap technique is important because deviations from normality can be skewed or heavy-tailed distributions. Therefore, from the interval estimation of a population percentile and by taking the variation of the Gumbel distribution, we determined Equation (15) as follows:

$$p\left\{\hat{\mu} + u_p\hat{\sigma} - t_\gamma\sigma\sqrt{\frac{1}{n} + u_p^2(\beta^2 - 1)} < \mu + u_p\sigma\right\} = \gamma , \tag{15}$$

Moreover, the interval estimation should be limited from the lower confidence of one side and the upper confidence on the other side of natural logarithmic fatigue life, which is relevant to the confidence level of $\gamma$ and the reliability of $p$, as written in Equations (16) and (17), respectively.

$$x_{p\gamma} = \ln N_{p\gamma} = \left( \hat{\mu} + u_p \hat{\sigma} - t_\gamma \sigma \sqrt{\frac{1}{n} + u_p^2 (\beta^2 - 1)} \right), \tag{16}$$

$$x_{p\gamma} = \ln N_{p\gamma} = \left( \hat{\mu} + u_p \hat{\sigma} + t_\gamma \sigma \sqrt{\frac{1}{n} + u_p^2 (\beta^2 - 1)} \right), \tag{17}$$

The amount of Equation (18) represents the probabilistic value of fatigue life prediction that can be integrated into any fatigue life model. Moreover, this value exhibits a random factor of unforeseen perturbation in the deterministic model. Furthermore, it is a natural log Gumbel stochastic process with zero $\hat{\mu}$, a constant $\sigma$, and a non-negative value.

$$Z(N) = u_p \hat{\sigma} - t_\gamma \sigma \sqrt{\frac{1}{n} + u_p^2 (\beta^2 - 1)} = \hat{\sigma} \left( u_p - t_\gamma \sqrt{\frac{1}{n\beta^2} + u_p^2 \left( 1 - \frac{1}{\beta^2} \right)} \right), \tag{18}$$

The total strain life includes elastic and plastic amplitude parts. Therefore, a model was introduced based on the statistical approach by using the strain fatigue life relationship. The idea was to integrate the probabilistic model parts into the traditional strain fatigue life, such as the Coffin–Manson, Morrow, and SWT models. A random factor (random disturbance) $Z(N)$, which was explained in Equation (18), was added to the deterministic model to determine the random nature of fatigue life. This random disturbance interacted separately with each elastic and plastic part of the Coffin–Manson strain life model (Equation (19)), as written in Equation (20).

$$\varepsilon_a = \frac{\sigma_f'}{E}(2N_f)^{b_1} + \varepsilon_f'(2N_f)^{b_2} = \varepsilon_e + \varepsilon_p \,, \tag{19}$$

$$\begin{cases} \ln \varepsilon_e = \ln\left( \frac{\sigma_f'}{E}(2N_f)^{b_1} \right) + Z(N) \\ \ln \varepsilon_p = \ln\left( \varepsilon_f'(2N_f)^{b_2} \right) + Z(N) \end{cases}, \tag{20}$$

where $\varepsilon_a$, $\sigma_f'$, $\varepsilon_f'$, $E$, $b_1$, and $b_2$ are the total strain amplitude, fatigue strength coefficient, fatigue ductility coefficient, modulus of elasticity, fatigue strength exponent, and fatigue ductility exponent, respectively. The random variables can be written in the form of dimensionless ($\widetilde{N}_f = \frac{N_f}{N_0}$) to characterize the relationship between the initial variables. Thus, Equation (20) can be written as follows:

$$\begin{cases} \ln \varepsilon_e = \ln\left( \frac{\sigma_f'}{E} \right) + \ln\left( 2\frac{N_f}{N_0} \right)^{b_1} + Z(N) \rightarrow \varepsilon_e = \left( \frac{\sigma_f'}{E} \right)\left( 2\widetilde{N}_f \right)^{b_1} . exp(Z(N)) \\ \ln \varepsilon_p = \ln\left( \varepsilon_f' \right) + \ln\left( 2\frac{N_f}{N_0} \right)^{b_2} + Z(N) \rightarrow \varepsilon_p = \left( \varepsilon_f' \right)\left( 2\widetilde{N}_f \right)^{b_2} . exp(Z(N)) \end{cases}, \tag{21}$$

The probabilistic Coffin–Manson strain fatigue life model was obtained by substituting Equation (18) with (21) and then substituting it with Equation (19). Hence, the newly proposed probabilistic modeling equations for the Coffin–Manson, Morrow, and SWT strain fatigue life models can be modeled as follows:

Coffin–Manson:

$$\varepsilon_a = \left( \left( \frac{\sigma_f'}{E} \right)\left( 2\widetilde{N}_f \right)^{b_1} + \left( \varepsilon_f' \right)\left( 2\widetilde{N}_f \right)^{b_2} \right) . exp\left( \hat{\sigma} \left( u_p - t_\gamma \sqrt{\frac{1}{n\beta^2} + u_p^2 \left( 1 - \frac{1}{\beta^2} \right)} \right) \right), \tag{22}$$

Morrow:

$$\varepsilon_a = \left(\left(\left(\frac{\acute{\sigma}_f - \sigma_m}{E}\right)\left(2\widetilde{N}_f\right)^{b_1} + \acute{\varepsilon}_f\left(\frac{\acute{\sigma}_f - \sigma_m}{\acute{\sigma}_f}\right)^{\frac{b_2}{b_1}}\left(2\widetilde{N}_f\right)^{b_2}\right).exp\left(\hat{\sigma}\left(u_p - t_\gamma\sqrt{\frac{1}{n\beta^2} + u_p^2\left(1 - \frac{1}{\beta^2}\right)}\right)\right)\right), \quad (23)$$

SWT:

$$\sigma_{max}.\varepsilon_a = \left(\left(\left(\frac{\left(\acute{\sigma}_f\right)^2}{E}\right)\left(2\widetilde{N}_f\right)^{2b_1} + \left(\acute{\sigma}_f\acute{\varepsilon}_f\right)\left(2\widetilde{N}_f\right)^{b_1+b_2}\right).exp\left(\hat{\sigma}\left(u_p - t_\gamma\sqrt{\frac{1}{n\beta^2} + u_p^2\left(1 - \frac{1}{\beta^2}\right)}\right)\right)\right), \quad (24)$$

where $\varepsilon_a$, $\acute{\sigma}_f$, $\acute{\varepsilon}_f$, $E$, $b_1$, $b_2$, and $\sigma_m$ are the total strain amplitude, fatigue strength coefficient, fatigue ductility coefficient, modulus of elasticity, fatigue strength exponent, fatigue ductility exponent, and mean stress, respectively. Consequently, the advantage of this procedure is that it can likewise be applied to a stress fatigue life model.

## 3. Results and Discussion

### 3.1. Measured Strain Time History Signals

Collected strain signals were used to analyze the fatigue life of vehicle components, particularly a coil spring. In this case, strain-time history signals were measured from a vehicle coil spring for rural, campus, and highway road excitations, as depicted in Figure 6. Each strain signal included variable amplitude from road profile roughness [37]. Therefore, the strain gauge was adjusted on the basis of the mechanism of an automobile suspension system using a data acquisition system.

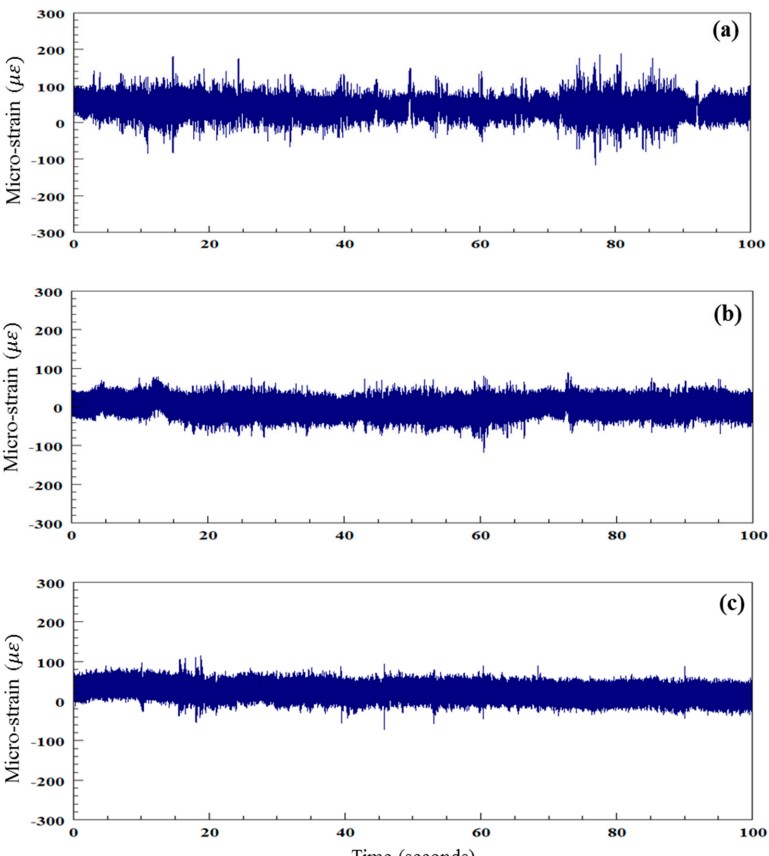

**Figure 6.** Strain signals measured for (**a**) rural, (**b**) campus, and (**c**) highway road loads.

### 3.2. Approximate Best Model Compared with Other Models

The AIC method was considered in this section to estimate the best approximate model from several models. The AIC method is commonly used to evaluate different models given a specific data set [38]. Accordingly, the Gumbel, Gamma, and Gaussian models were compared to obtain the best approximation model. In general, the AIC formulation is assumed when the sample size is adequately large, but if the division of the sample size over the number of parameters is quite small ( $\frac{n}{k} < \sim 40$ ), then the AIC is formulated as follows [39,40]:

$$AIC = -2(Log - Likelihood) + 2k + \left[ \frac{2k(k+1)}{n-k-1} \right], \tag{25}$$

where $k$ and $n$ are the number of parameters and the sample size, respectively. In this respect, the lowest value of the AIC method is the best estimation; thus, the Gumbel model illustrated the best appropriate parameters of 314.78 compared with the Gamma and Gaussian distributions, as presented in Table 2.

**Table 2.** Evaluation of different probabilistic models based on Akaike information criterion method.

| Method | Model ($k = 2$ and $n = 9$) | | |
|---|---|---|---|
| | Gumbel | Gamma | Gaussian |
| Akaike information criterion | 314.78 | 1041.79 | $3.04 \times 10^{13}$ |
| Log–likelihood | −154.39 | −517.89 | $-1.53 \times 10^{13}$ |

### 3.3. Strain Fatigue Prediction based on Rural, Campus, and Highway Road Excitatuions

Fatigue life predictions based on three road excitations were obtained according to the strain-time histories illustrated in Figure 6 and results are presented in Table 3. Fatigue life was assessed using the Coffin–Manson, Morrow, and SWT strain life models for a vehicle coil spring suspension system. The fatigue life prediction value of the coil spring was in the range of $2 \times 10^4$ to $3 \times 10^5$ cycles. The Coffin–Manson model presented the highest and lowest fatigue life and damage values for the three roads, but the mean strain or stress was not presented in the Coffin–Manson model. For mechanical components, such as the coil spring, the mean strain or stress indicated a considerable effect on long lives when the components were subjected to vibration loadings [41]. Kamaya and Kawakubo [42] presented that mean strain or stress can be applied to the components to shorten fatigue life.

**Table 3.** Fatigue life and damage values for different road surfaces.

| Strain Life Model | Road Surface | Fatigue Life (Block Cycle) | Fatigue Damage (1/cycle) |
|---|---|---|---|
| Coffin–Manson | Rural | $2 \times 10^5$ | $5 \times 10^{-6}$ |
| | Campus | $2 \times 10^5$ | $5 \times 10^{-6}$ |
| | Highway | $3 \times 10^5$ | $3 \times 10^{-6}$ |
| Morrow | Rural | $3 \times 10^4$ | $3 \times 10^{-5}$ |
| | Campus | $7 \times 10^4$ | $1 \times 10^{-5}$ |
| | Highway | $2 \times 10^5$ | $5 \times 10^{-6}$ |
| Smith–Watson–Topper | Rural | $2 \times 10^4$ | $5 \times 10^{-5}$ |
| | Campus | $3 \times 10^4$ | $3 \times 10^{-5}$ |
| | Highway | $1 \times 10^5$ | $1 \times 10^{-5}$ |

### 3.4. Proposed Probabilistic Method Based on the Gumbel Model

A probabilistic model was proposed based on strain fatigue life data. According to this deterministic data, several variables can be changed by using random variables. Therefore, a

statistical distribution should be characterized by each of the random variables when strain fatigue life data are used owing to experimental assumptions. In addition, the proposed probabilistic method was considered to correlate the strain fatigue life data.

In this step of the proposed probabilistic method, the parameters of the Gumbel distribution should be evaluated to derive probabilistic $\varepsilon$-$N$ when the model is a non-normal distribution. Therefore, this value can be obtained based on the bootstrap method for the Gumbel distribution model. The bootstrap method determines deviations from normality that are small and skewed, as illustrated in Figure 7. The bootstrap technique is mostly used to infer statistical random data based on simulations. The main goal of this method is to resample random data from an original set that produced a large number of repeated random datasets to approximate the statistics of the sampling distribution, as depicted in Figure 7 [43]. In this research, life data were considered as sample size ($n = 9$) based on the strain approach from rural, campus, and highway road excitations. The 95% confidence interval for $\mu$ was considered, and 1000 bootstrap samples were selected to ensure the central limit theorem in order to obtain $t$-distribution values of the bootstrap method (Figure 8). Therefore, $t_\gamma$ was estimated for the lower case ($t_{0.025} = -1.8014$) and the upper case ($t_{0.975} = 1.818$) based on 0.025 percentiles. The estimated parameters based on MLE for the Gumbel model for the probabilistic mathematical model (Equation (18)) are listed in Table 4. Furthermore, the correction coefficient $\beta$ values of standard deviations estimated for rural, campus, and highway road profiles according to the Coffin–Manson, Morrow, and SWT strain life models were 47.56, 47.95, and 43.98, respectively.

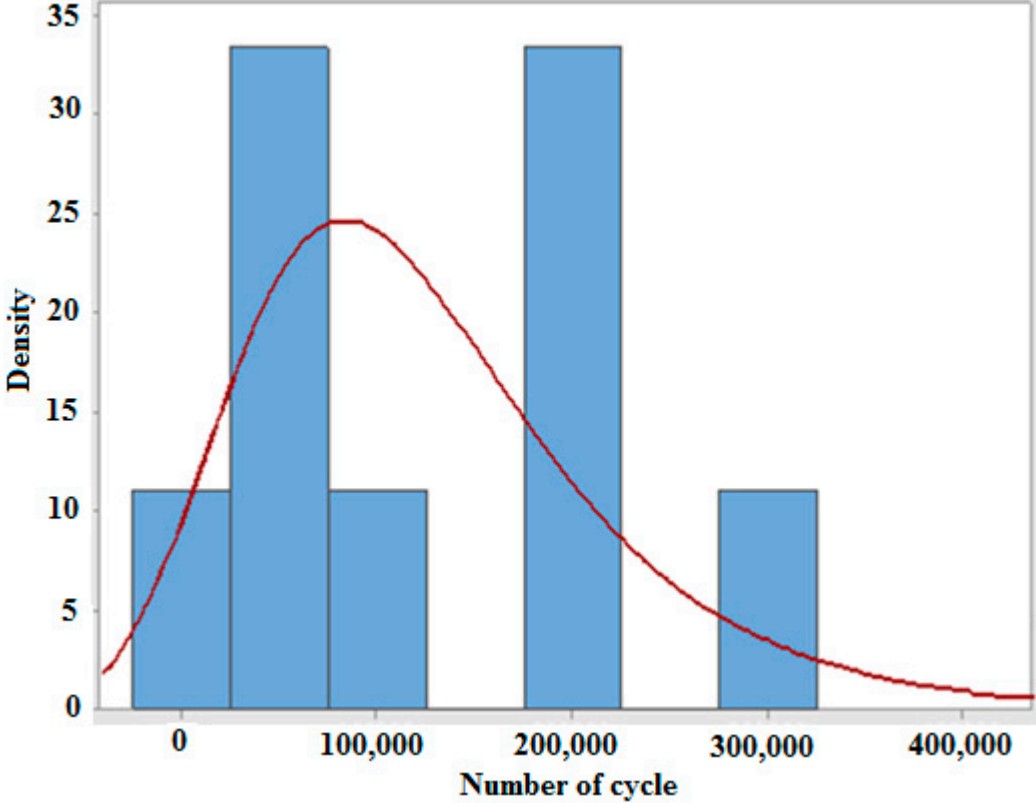

**Figure 7.** Probability density function of the Gumbel distribution for a lightly skewed distribution.

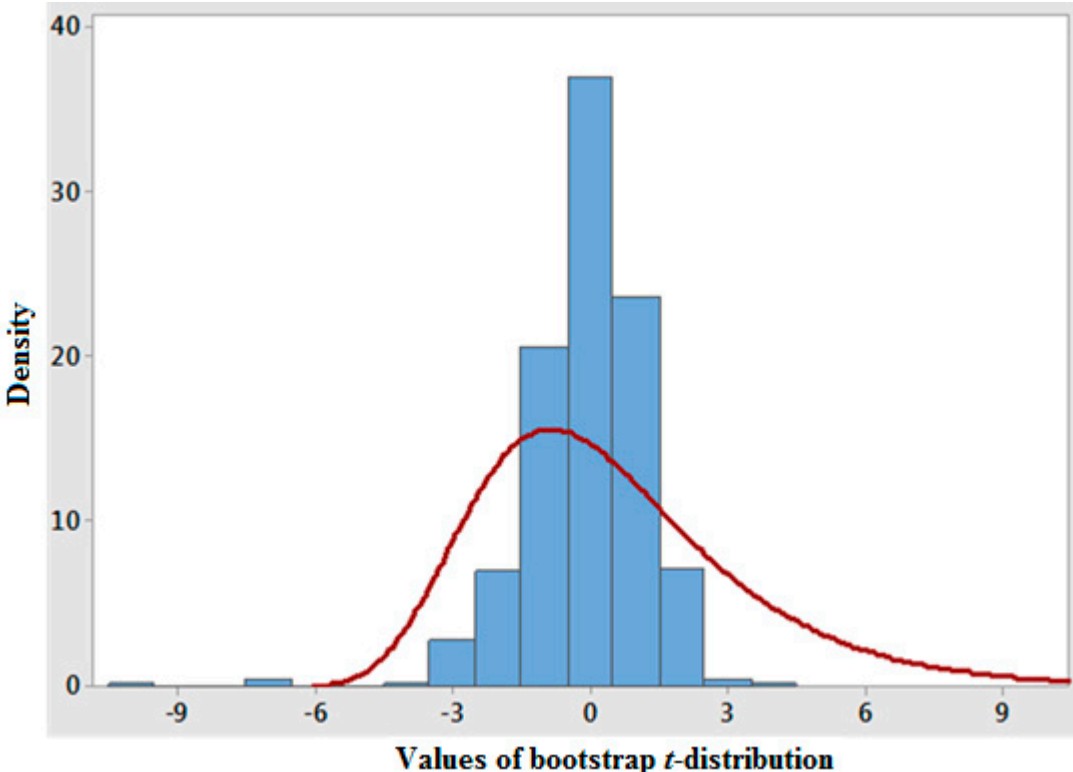

**Figure 8.** PDF of the Gumbel distribution for *t*-distribution values of the bootstrap method.

**Table 4.** Estimated parameters for the Gumbel distribution using maximum likelihood estimation.

| Fatigue Life Model | Unbiased Scale Parameter ($\hat{\sigma}$) | Unbiased Location Parameter ($\hat{\mu}$) | $u_p$ |
| --- | --- | --- | --- |
| Coffin–Manson | 39,526 | 259,524 | 0.0165 |
| Morrow | 58,299 | 124,304 | 0.0436 |
| Smith–Watson–Topper | 37,505 | 72,104 | 0.0444 |

The proposed model was evaluated by the experimental data of Goncalves et al. [44], which was obtained for SAE 5160 carbon steel, and then plotted for comparison. The probabilistic model was applied to 9 lives of the strain fatigue life models (each rural, campus, and highway road load was allocated 3 lives based on the Coffin–Manson, Morrow, and SWT models, separately), as shown in Table 3. The best fit parameters based on MLE were performed according to the Gumbel distribution model (Table 4). The results of the fitted parameters using the proposed strain-based probabilistic mathematical model are shown in Figures 9–11. Moreover, these figures demonstrated a reasonable fit compared with their measured strain fatigue life curves and experimental data when the probabilistic model was applied. In this regard, root-mean-square error (RMSE) calculations were performed in logarithmic scale to determine the accuracy with the proposed model and experimental data based on the Coffin–Manson, Morrow, and SWT strain life models (Figures 12–14). In addition, the prediction errors were used to evaluate the accuracy of the fatigue life prediction models based on the log–log scale of 1:2, 1:1, and 2:1 fatigue correlation for the Coffin–Manson, Morrow, and SWT strain fatigue life models, respectively, between the measured strain and proposed probabilistic fatigue life predictions. The Coffin–Manson strain life model demonstrated a high correlation (approximately 0.00114) with the proposed model and experimental data using the RMSE method (Figure 12). The Coffin–Manson model exhibited a maximum strain amplitude and a strain endurance limit of approximately $\varepsilon_a = 0.3329$ and $\varepsilon_{a_0} = 0.0037$, respectively. The probabilistic method was considered, and the maximum strain amplitude and strain endurance limit decreased to approximately $\varepsilon_a = 0.2599$ and $\varepsilon_{a_0} = 0.0029$,

respectively. Figure 9 presents the experimental data confirming that the proposed probabilistic curve was conservative for the Coffin–Manson strain fatigue life model.

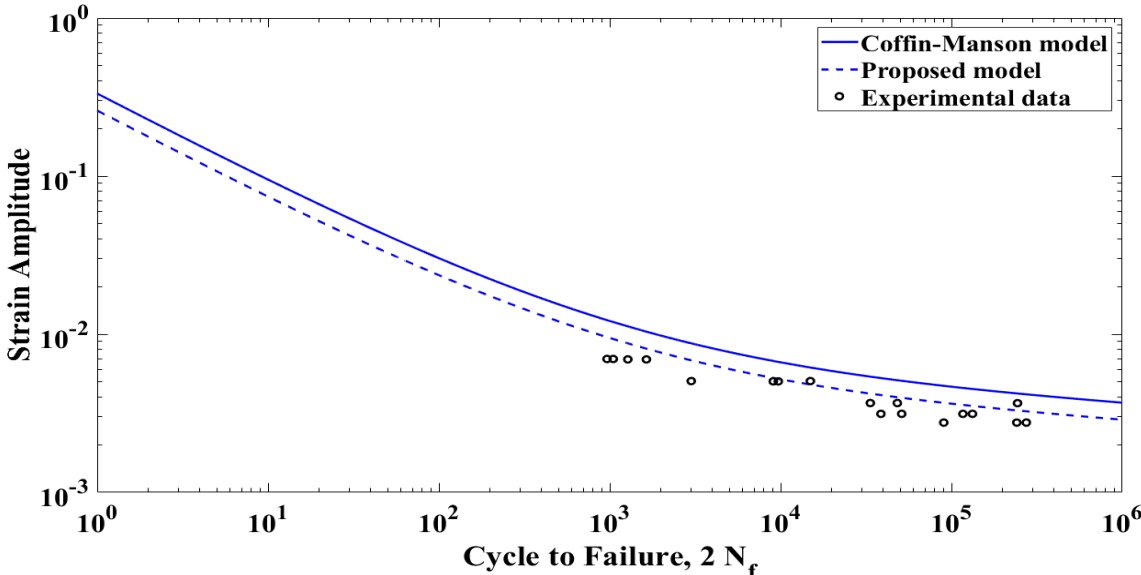

**Figure 9.** The Coffin–Manson and proposed probabilistic models using the Gumbel distribution for rural, campus, and highway road excitations.

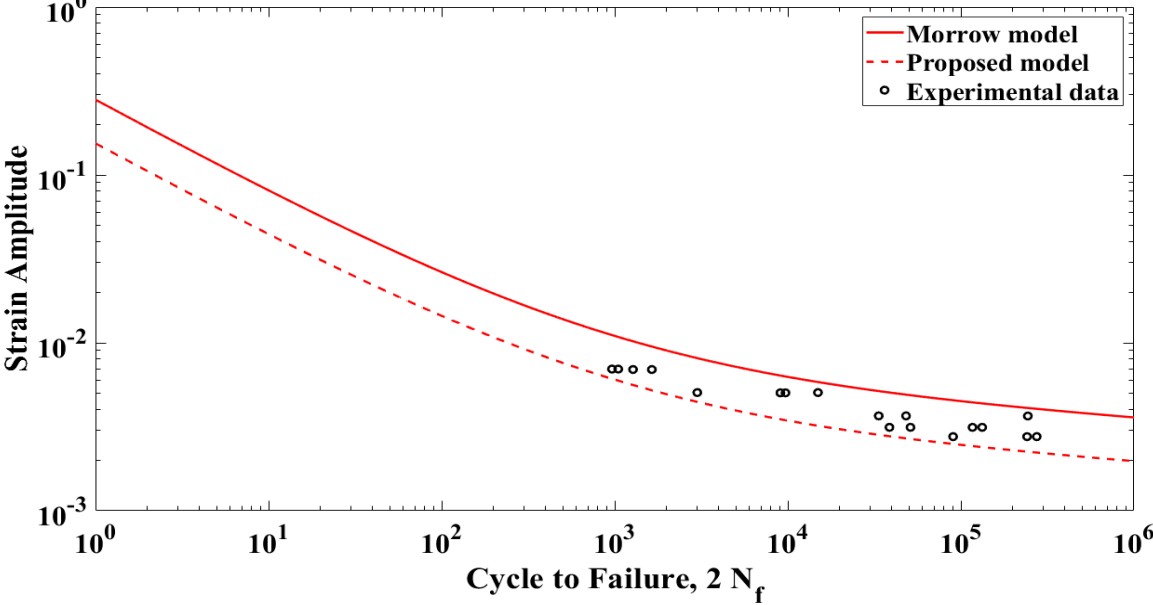

**Figure 10.** The Morrow and proposed probabilistic models using the Gumbel distribution for rural, campus, and highway road excitations.

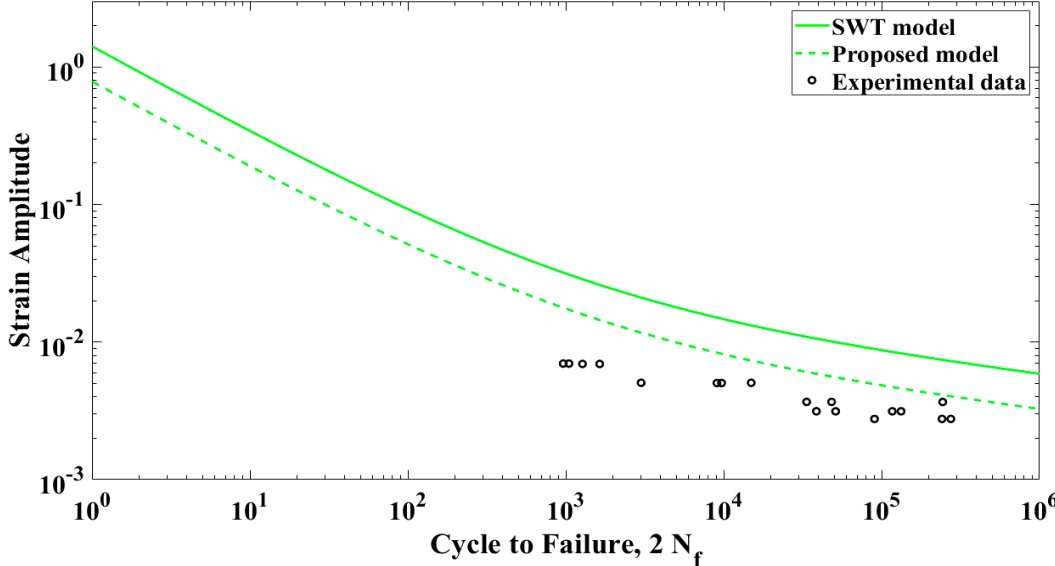

**Figure 11.** The Smith–Watson–Topper and proposed probabilistic models using the Gumbel distribution for rural, campus, and highway road excitations.

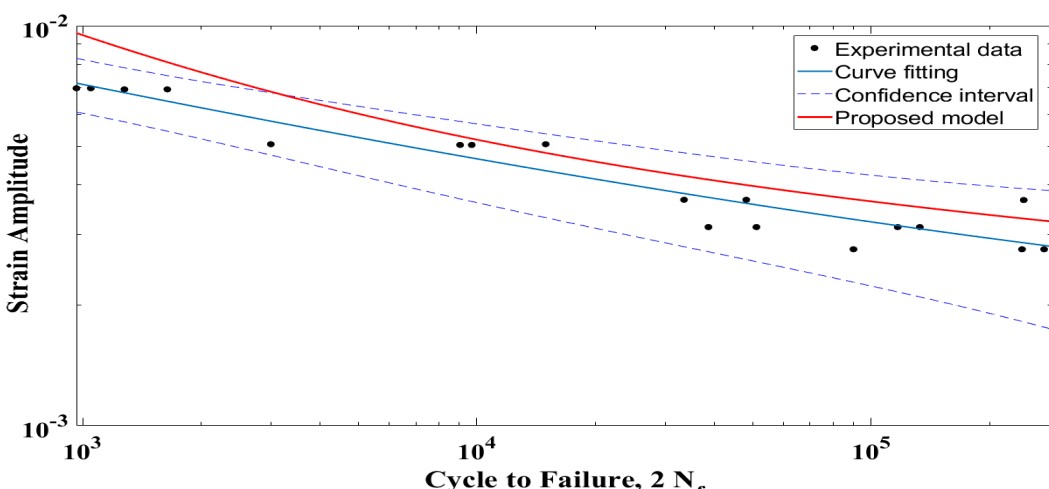

**Figure 12.** Correlation with the proposed model and experimental data based on the Coffin–Manson strain–life model.

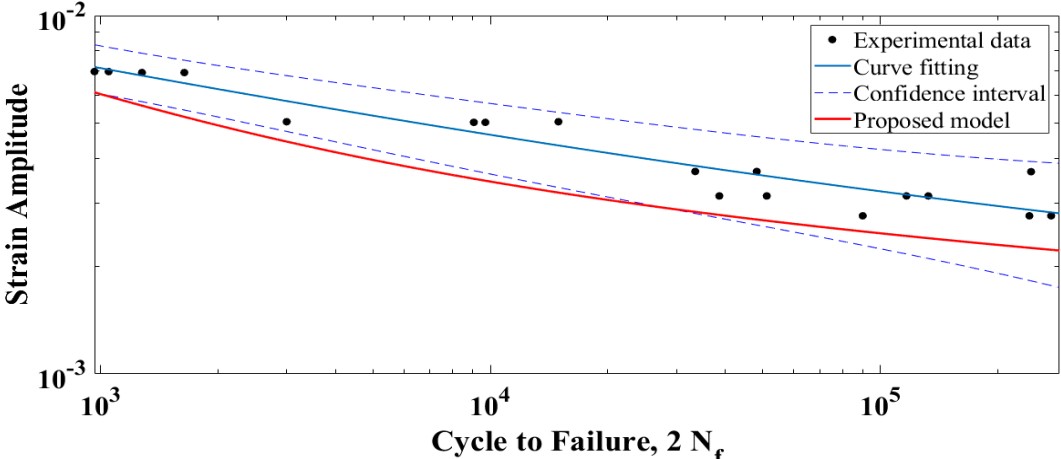

**Figure 13.** Correlation with the proposed model and experimental data based on the Morrow strain-life model.

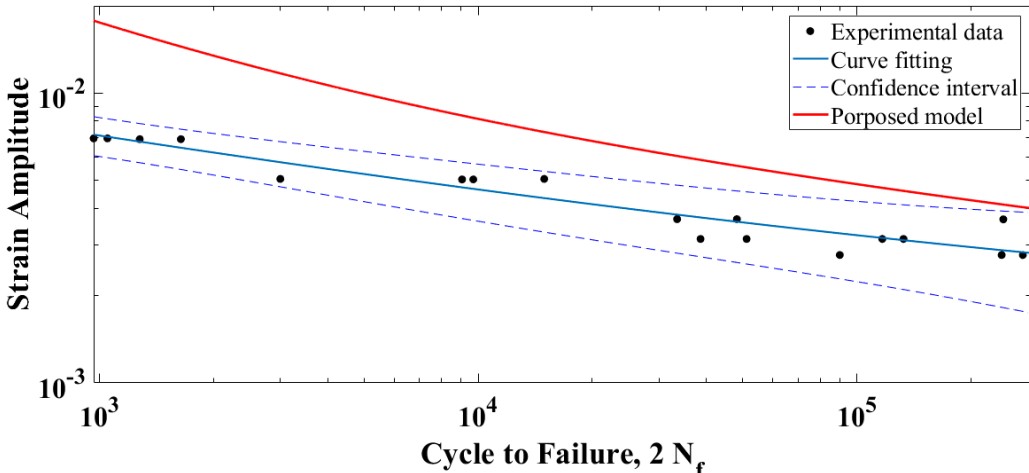

**Figure 14.** Correlation with the proposed model and experimental data based on the Smith–Watson–Topper strain-life model.

The approach used for the Coffin–Manson model was considered to obtain the best fit parameters for the Morrow and SWT strain life models (Table 4). According to Table 5, the obtained mean stress effect $\sigma_m$ was 42 MPa using the Ramberg–Osgood relationship. This value was then applied to the Morrow strain life model on the basis of rural, campus, and highway strain-time history signals. The Morrow model was applied in order to calculate mean stress effect where components were subjected to compressive and tensile loadings [42]. Therefore, the proposed probabilistic model illustrated non-conservative fatigue life prediction based on the experimental data when the mean stress was presented (Figure 10). Mean stress is significantly affected when the elastic strain amplitudes are dominant [41]. Thus, the Morrow model predicts fatigue lives based on experimental data beyond the strain life curve where vehicle coil spring is subjected to tensile mean stress. In the case of the SWT model, the product of $\sigma_{max}$ and $\varepsilon_a$ parameters remained constant for various combinations of maximum stress and strain amplitude. Thus, the SWT model predicts good assessment, where mean stress effect is applied under tensile mean stress cases in the low-cycle fatigue approach [45]. The maximum stress $\sigma_{max}$ obtained was 433 MPa to investigate the SWT strain life model. Therefore, the proposed probabilistic model presented a good agreement and a safe life with the experimental data for the SWT model (Figure 11). The maximum strain amplitude and the strain endurance limit for the Morrow strain curve were approximately $\varepsilon_a = 0.2808$ and $\varepsilon_{a_0} = 0.0036$, respectively. These values were approximately $\varepsilon_a = 0.1542$ and $\varepsilon_{a_0} = 0.002$ using the probabilistic of the mathematical model. In addition, the RMSE values based on the Morrow and SWT strain life models were approximately 0.00107 and 0.00509, respectively, indicating a good correlation with the proposed model and experimental data (Figures 13 and 14). With the SWT strain life model, the maximum strain amplitude and the strain endurance limit were determined to be approximately $\varepsilon_a = 1.4128$ and $\varepsilon_{a_0} = 0.0059$ for the SWT model and $\varepsilon_a = 0.7845$ and $\varepsilon_{a_0} = 0.0033$ for the proposed probabilistic model, respectively. Table 6 shows a summary of results of the three strain life models according to above explanation. The most obvious findings to emerge from the analysis were that the deterministic formulas of the Coffin–Manson, Morrow, and SWT strain life models showed a good agreement or safe life with the experimental data. The results of the application of the proposed probabilistic model were conservative, non-conservative, and safe for the Coffin–Manson, Morrow, and SWT strain life models, respectively.

**Table 5.** Variable amplitudes of measured strain signals for different road surfaces.

| Road Surface | Maximum Strain ($\mu\varepsilon$) | Minimum Strain ($\mu\varepsilon$) | Mean strain ($\mu\varepsilon$) |
|---|---|---|---|
| Rural | 189 | −116 | 39 |
| Campus | 87 | −118 | −0.47 |
| Highway | 114 | −72 | 22 |

**Table 6.** Different values of maximum strain amplitude and endurance limit.

| Fatigue Life Model | Measured Strain Fatigue Life Curve | | Proposed Probabilistic Fatigue Life Curve | | |
|---|---|---|---|---|---|
| | ($\varepsilon_a$) | ($\varepsilon_{a_0}$) | ($\varepsilon_a$) | ($\varepsilon_{a_0}$) | RMSE |
| Coffin–Manson | 0.3329 | 0.0037 | 0.2599 | 0.0029 | 0.00114 |
| Morrow | 0.2808 | 0.0036 | 0.1542 | 0.002 | 0.00107 |
| Smith–Watson–Topper | 1.4128 | 0.0059 | 0.7845 | 0.0033 | 0.00509 |

These results may be explained by the fact that the mean stress effect showed a tendency to overestimate low-cycle fatigue for the Morrow model. The product of maximum stress ($\sigma_{max}$) and strain amplitude ($\varepsilon_a$) can influence the mean stress and the strain amplitude for the SWT model. Although Equation (24) demonstrated a satisfactory relationship in the presence of mean stress in the high-cycle fatigue regime, it was conservative in short lives. Nevertheless, the mean stress can change the relevance between the elastic and plastic strain amplitudes of the Morrow model indirectly, whereas the SWT model may not be conservative when the mean stress effect was compressive [45,46].

*3.5. Validation of Fatigue Life Prediction*

The predicted strain fatigue life models using the Coffin–Manson, Morrow, and SWT models were plotted against the proposed probabilistic fatigue life model. In this case, fatigue lives were plotted based on log–log scales of 1:2 or 2:1 fatigue correlation and $R^2$ curves to evaluate the accuracy of the fatigue life prediction models [47,48] (Figures 15–20). Therefore, predicted strain fatigue lives indicated below 1:2 represented a conservative prediction, whereas those above denoted non-conservative predictions (Figures 15, 17 and 19). The correlation curve showed a good correlation within the range of 1:2 and 2:1 lines for the Coffin–Manson strain fatigue life model between the measured strain and proposed probabilistic fatigue life predictions (Figure 15). Furthermore, the $R^2$ curve demonstrated a good agreement value of approximately 0.9971 for the Coffin–Manson strain fatigue life model (Figure 16). The approach used for the Morrow and SWT strain life models was considered and performed. Figures 17 and 19 show a good correlation between the measured strain and the proposed probabilistic fatigue life predictions. Several data values demonstrated that the predictions became conservative for the Morrow and SWT strain life models. In addition, the $R^2$ values exhibited a good agreement for the two strain life models of approximately 0.9833 and 0.9962 (Figures 18 and 20). In the meantime, some data from Figures 15, 17 and 19 were indicated in beyond the boundary conditions because the results of fatigue life prediction between the measured strain life and proposed probabilistic models were not in the same range.

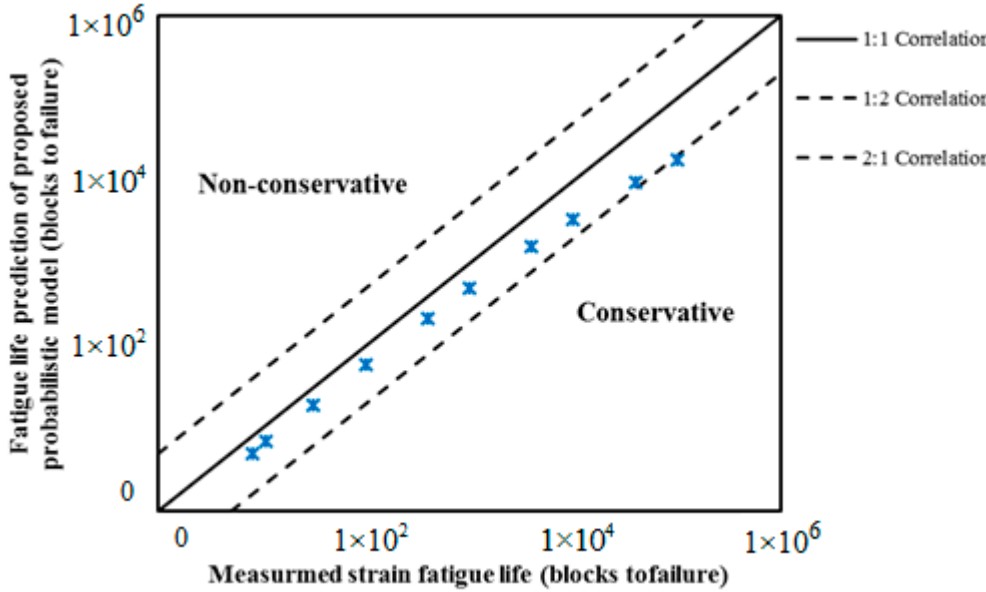

**Figure 15.** Fatigue life correlation based on the Coffin–Manson strain-life model.

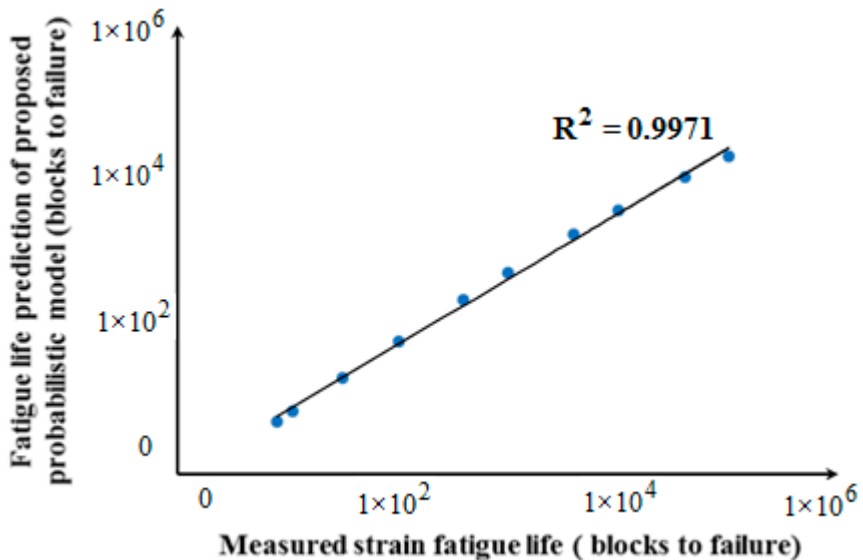

**Figure 16.** $R^2$ accuracy curve based on the Coffin–Manson strain-life model.

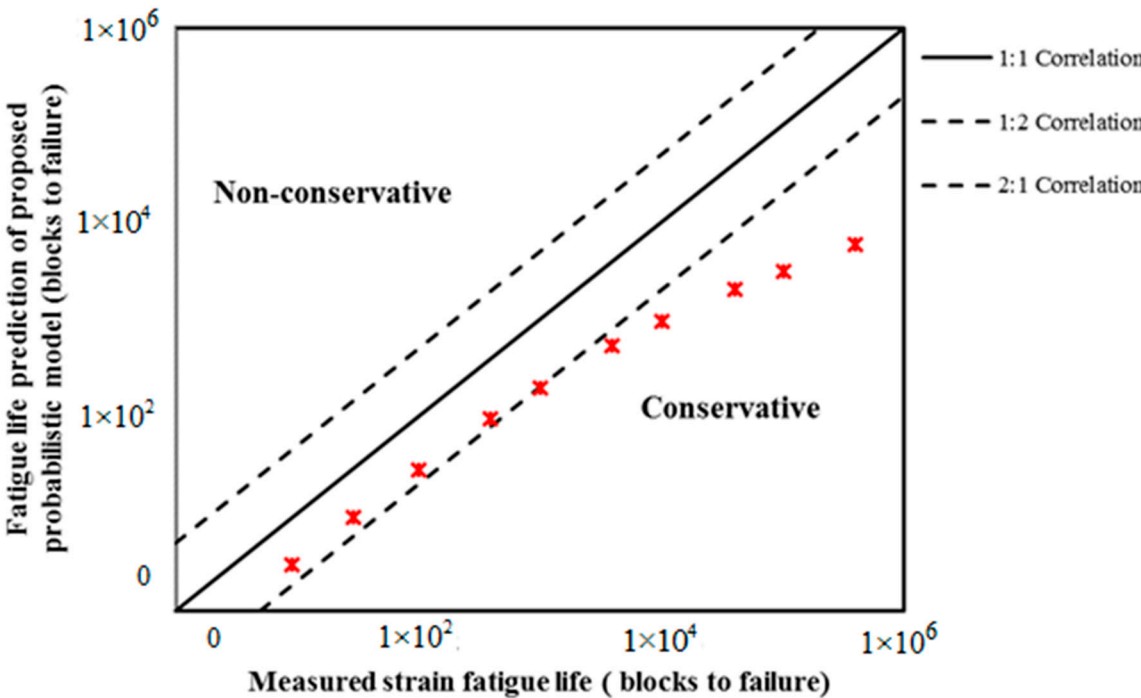

**Figure 17.** Fatigue life correlation based on the Morrow strain-life model.

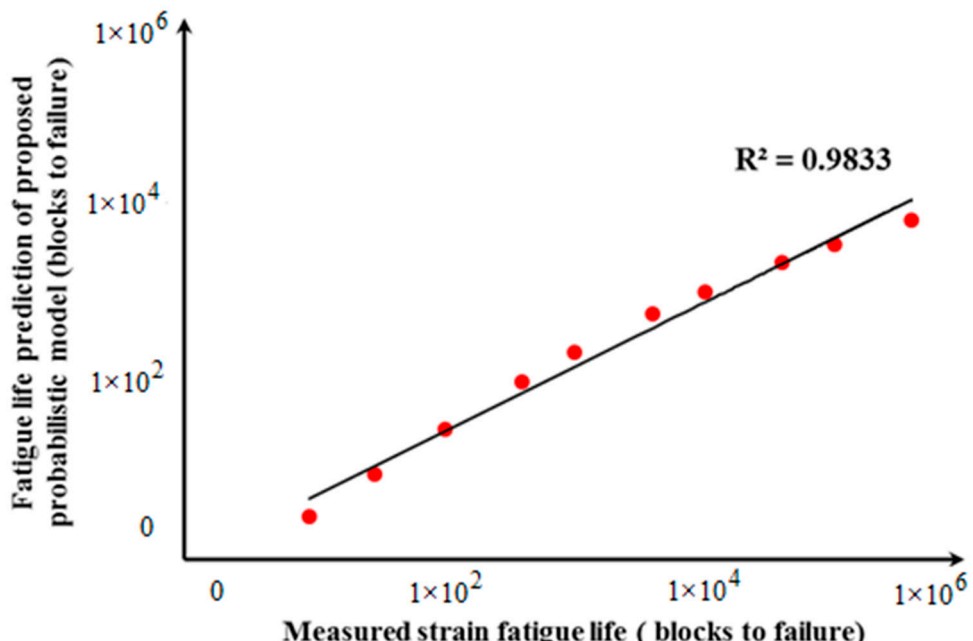

**Figure 18.** $R^2$ accuracy curve based on the Morrow strain-life model.

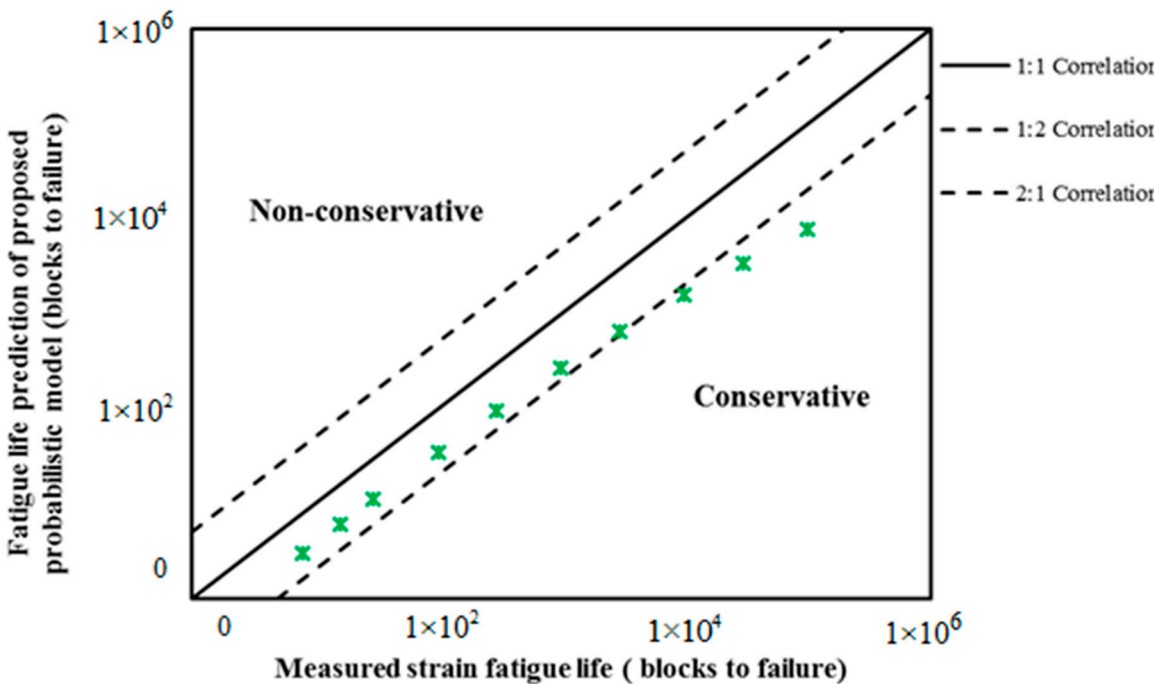

**Figure 19.** Fatigue life correlation based on the Smith–Watson–Topper strain-life model.

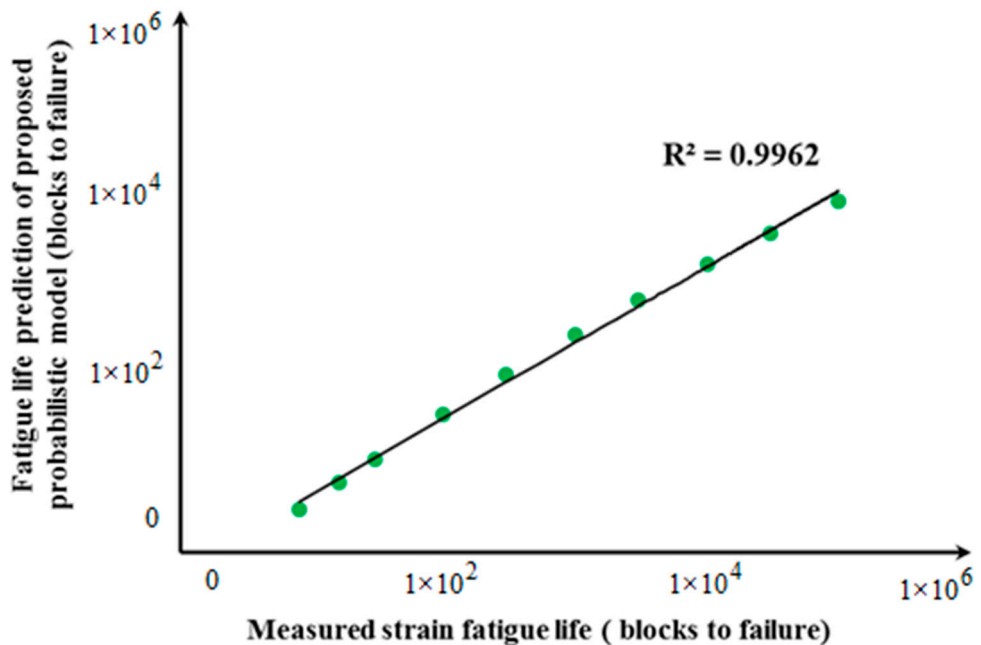

**Figure 20.** $R^2$ accuracy curve based on the Smith–Watson–Topper strain-life model.

In most statistics data, another possible correlation of the goodness-of-fit test is the PCC technique, which can evaluate within the range of $-1 \leq r \leq 1$, where $-1$ and 1 values indicate thorough negative and positive linear correlations, respectively. The zero value illustrates the absence of a linear correlation. The results of the predicted statistical fatigue life measured the strength of the linear correlation between the measured strain and the proposed probabilistic fatigue life, which is denoted by $r$. Therefore, the Coffin–Manson, SWT, and Morrow strain life models showed a highly potent PCC value of approximately $r = 0.991$, $r = 0.988$, and $r = 0.944$, respectively (Figure 21).

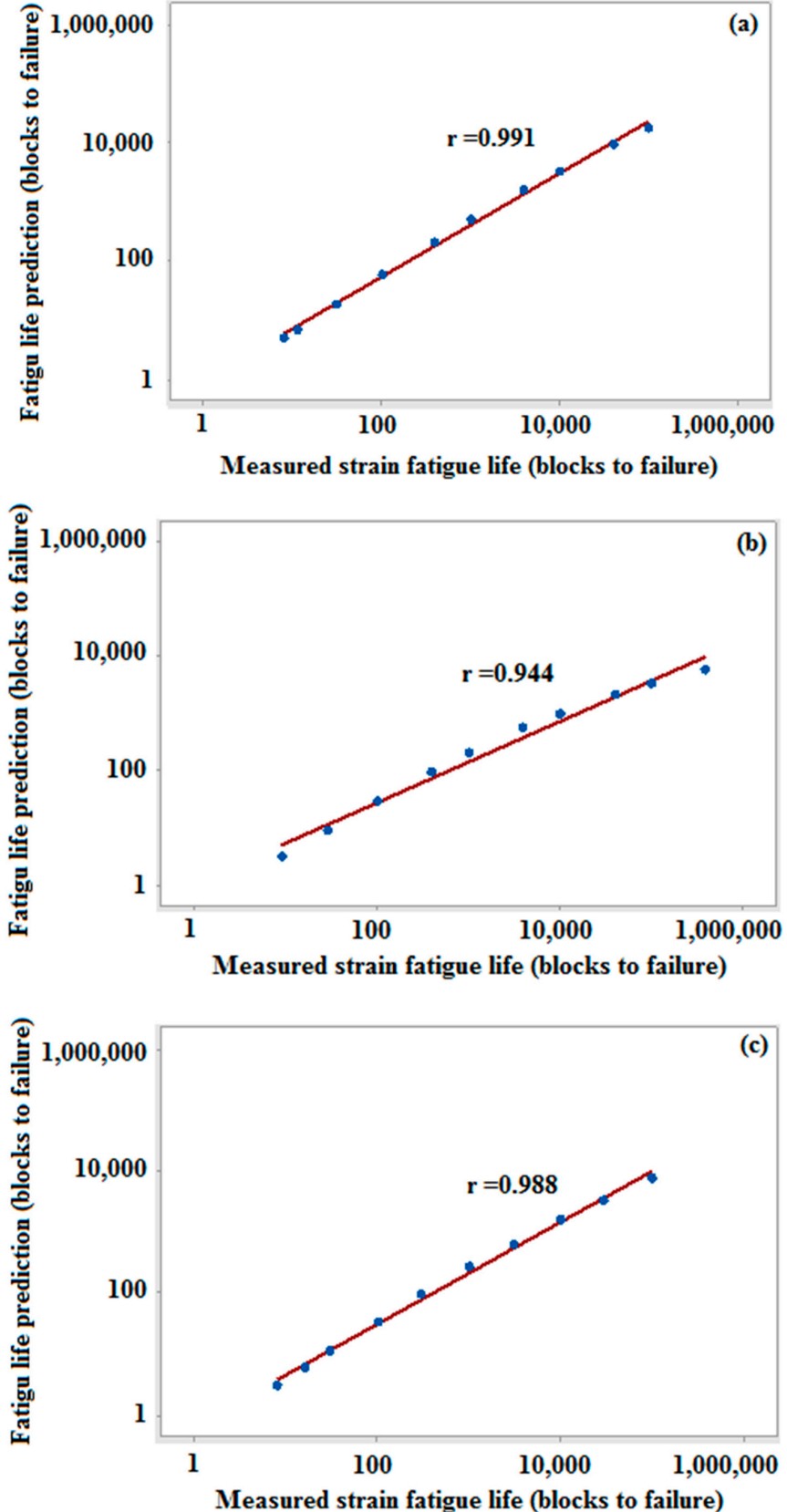

**Figure 21.** Pearson correction coefficient for (**a**) Coffin–Manson, (**b**) Morrow, and (**c**) Smith–Watson–Topper models.

## 4. Conclusions

This study proposed a framework to explore a mathematical model based on the probabilistic method by using a probabilistic strain-life ($\varepsilon$-$N$) Gumbel distribution model. Fatigue life prediction data were applied to distinguish the statistical Gumbel strain-life model to determine probabilistic strain life curves for an automobile coil spring. These curves were established using an $\varepsilon$-$N$ field for the Coffin–Manson, Morrow, and SWT strain life models on the basis of rural, campus, and highway road excitations. The strain-time history measurements of these different road excitations were considered as an overall signal to inspect the different responses of a coil spring when a vehicle passes through the various road surfaces.

The PDF of the Gumbel distribution was considered for the proposed probabilistic mathematical model to estimate parameters by using the MLE method. Moreover, each strain life model showed that the proposed probabilistic model can be the best model for the deterministic fatigue data when integrated with the other models. This finding determined that the presence of mean and maximum stress can considerably affect fatigue life prediction for low- and high-cycle regimes. The prediction errors were estimated in a logarithmic scale based on RMSE to determine the accuracy of correlations with the proposed model and experimental data. Moreover, the prediction errors were used to evaluate the accuracy of the fatigue life prediction models based on log–log scales of 1:2, 1:1, and 2:1 fatigue correlation for the Coffin–Manson, Morrow, and SWT strain fatigue life models between the measured strain and proposed probabilistic fatigue life predictions, respectively. Therefore, the RMSE values based on the Coffin–Manson, Morrow, and SWT strain life models were approximately 0.00114, 0.00107, and 0.00509, respectively, which showed a good correlation with the proposed model and experimental data. Moreover, the $R^2$ and PCC methods were used to evaluate fatigue life prediction between the measured strain fatigue life and the proposed probabilistic model. In this regard, the Coffin–Manson model showed a highly potent agreement with the $R^2$ and PCC values of approximately 0.9971 and $r = 0.991$ when a zero mean stress effect was presented. Furthermore, the Morrow and SWT models demonstrated a good agreement of approximately 0.9833 and 0.9962 based on $R^2$ values and $r = 0.944$ and $r = 0.988$ based on PCC amounts, respectively. Finally, the proposed mathematical model demonstrated potential in strain fatigue life prediction models using the Gumbel distribution. Automobile suspension components, however, should be surveyed further for reliability assessment and probabilistic fatigue data.

**Author Contributions:** In this work, R.M. conducted the analysis and writing under supervision of S.A. and S.S.K.S. provided guidance on manuscript writing. All authors have read and agreed to the published version of the manuscript.

**Funding:** The authors wish to acknowledge UKM research grants (FRGS/1/2019/TK03/UKM/01/3 and FRGS/1/2019/TK03/UKM/02/1) for the research funding.

**Conflicts of Interest:** The authors declare no conflict of interest.

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
