# Peer review of "Fatigue Reliability Assessment of an Automobile Coil Spring under Random Strain Loads Using Probabilistic Technique"

_metals, doi:10.3390/met10010012_

Round 1

Reviewer 1 Report

REVIEW

on the article

Reza Manouchehrynia, Shahrum Abdullah, and Salvinder Singh Karam Singh

 Probabilistic Modelling of Fatigue Reliability Prediction under Random Strain Loads

Summary. The article is devoted to the urgent issue of developing a mathematical model for predicting the reliability of coil springs under conditions of fatigue loading. Indeed, the vast majority of structural elements operate under conditions of accidental loading. The mechanical characteristics of steel used in the constructions also have a stochastic nature. Thus, not only the load is random, but the strength of the metal is stochastic.

The problem is complicated by the fact that the process of fatigue fracture of metals and alloys consists of two periods: the nucleation and propagation of fatigue cracks. And these periods are associated with various processes of changes in the structural and stress state of the material and the patterns of accumulation of damage during prolonged cyclic deformation. The duration of these fatigue failure periods depends on the level of strength and plastic characteristics of metallic materials.

            Thus, the article is devoted to the study of the problem of evaluating the reliability of the coil springs is relevant.

Comments.

However, the article has a lot of obscurities and a lot of inaccuracies. The title of the article is too extensive. The name implies that the authors developed a model that can describe the durability of all types of structures and materials: bridges, frames, trusses, buildings, and so on. As I already wrote, the fatigue life of a structure can be considered in several stages, namely the nucleation of a fatigue crack, the growth of a crack, and the loss of strength. Each stage is described by its laws, based on the structure and properties of the material. How is this taken into account in the model?

Specific comments.

 The article does not provide data on the material of the springs and their mechanical properties.  A crack originates on the surface. How was surface roughness taken into account in the model and surface preparation during testing?  How does the multistage development of a fatigue crack take into account the model?  Please, see .docx file. Figure 3 is also incorrect. Gumbel distribution from -∞ to +∞. Figure 4 is incomprehensible. What is the difference between curves a, b and c? Its authors estimated the prediction errors of the proposed model. Only not in logarithmic units, but in natural units. 

Author Response

Dear Reviewer,

Thank you for your useful comments on the structure of our manuscript. We have modified the manuscript accordingly, and the corrections are listed as follows. The imposed changes are in yellow color in the manuscript. We do appreciate the wise comment and try to satisfy each comment accordingly.

Question 1. The title of the article is too extensive.

Answer: Fatigue Reliability Assessment of an Automobile Coil Spring under Random Strain loads using Probabilistic Technique.

Question 2. The fatigue life of a structure can be considered in several stages, namely the nucleation of a fatigue crack, the growth of a crack, and the loss of strength. Each stage is described by its laws, based on the structure and properties of the material. How is this taken into account in the model?

Answer: In the case of fatigue life design, several stages were considered. The criteria of fatigue design have derived from infinite life to damage (defect) tolerance. Therefore, the criteria of fatigue design include utilization of four fatigue life models: (i) stress life (S-N) approach that cannot be used in durability analysis because this approach is included of two parts of crack initiation and crack propagation, (ii) strain life (ε-N) approach that has suitable usage in durability analysis, especially based on low cycle fatigue because it considers only fatigue crack initiation (nucleation) and the durability assessment of automotive components are carried out based on safe life, (iii) fatigue crack growth (da/dN-∆k) and (iv) two stage method (combination of section ii and iii) [25-27].

Therefore, according to above describe, this paper is taken into account only fatigue crack initiation (nucleation) because fatigue failure is correlated to the localize plasticity in low cycle fatigue. These expressions added to revise the manuscript (page 4, line 133 to 142).

SPECIFIC COMMENTS

Question 3. The article does not provide data on the material of the springs and their mechanical properties.

Answer: Thus, mechanical properties which are commonly used for coil spring SAE 5160 carbon steel is added in the revised version of the manuscript in Table 1 (Page 4).

Question 4.  A crack originates on the surface. How was surface roughness taken into account in the model and surface preparation during testing?

Answer: From Figure 2, coil spring surface is polished and scrubbed with a sand paper [23]. Therefore, the coil spring surface is smooth and there is no crack on the surface. A strain gauge was attached to the surface of a vehicle coil spring, then connected to a data logger system to analyse strain time history signals for three different road profiles. (Page 4, line 121 to 124).

Question 5. How does the multistage development of a fatigue crack take into account the model?

Answer: There is no crack on the coil spring surface and this paper discusses in crack initiation stage before failure based on strain control approach. Figure 3 illustrates a schematic of failure analysis based on total life. Hence, the research scope has been determined by safe life region (before failure).These expressions added to revise the manuscript (page 5, line 142 to 146).

Figure 3. Schematic of fatigue failure analysis. (Please, see the revised manuscript or .docx file (response to comments)).

Question 6. Equation (2) is already not probability function, because of  . In fact, the authors replaced x with an expression  and limited the numerical axis by ?.

Answer: Pleas, see the revised manuscript or .docx file (response to comments).

Question 7. Figure 3 is also incorrect. Gumbel distribution from -∞ to +∞.

Answer: A new figure added to the revised manuscript to show the schematic of the Gumbel probability density function (PDF) distribution (that is shown in revising manuscript by Figure 4). Therefore, this figure shows the limitation of the Gumbel distribution based on -∞ to +∞ (page 5, line 161 to 166).

In addition, Figure 5 represents the probability density curve of fatigue life. The vertical axis and horizontal axis in Figure 5 represent the probability of failure and fatigue life N, respectively.  These explanations are added and highlighted in the revised version of the manuscript (page 8, line 263 to 267).

Question 8. Figure 4 is incomprehensible. What is the difference between curves a, b and c?

Answer: The number of this figure is changed to Figure 6 and the caption of this figure has been changed to “Strain signals measured for (a) rural, (b) campus, (c) highway road loads (page 10). This Figure illustrates the different strain signals that are obtained based on the experimental test when vehicle is driven on different load profile. (Please, see the revised manuscript).

Question 9. Its authors estimated the prediction errors of the proposed model. Only not in logarithmic units, but in natural.

Answer: The prediction errors have been estimated in logarithmic scale based on root-mean-square-error (RMSE) to determine the accuracy with the proposed model and experimental data. In addition, the prediction errors have been used to evaluate the accuracy of the fatigue life prediction models based on log–log scales of 1:2, 1:1 and 2:1 fatigue correlation. These expressions added to revise the manuscript (page 13, line 422 to 428) and conclusion (Page 22, line 563 to 568).

Reviewer 2 Report

I accept after minor revisions, which include:

Line 11 - rewrite sentence

Line 317 - Figure 4 caption does not mention what is (a) (b) and (c).

Line 332 - why n=9? What constitutes a sample?

Tables 3 and 4 - remove unnecessary devimal places.

Author Response

Dear Reviewer,

Thank you for your useful comments on the structure of our manuscript. We have modified the manuscript accordingly, and the corrections are listed as follows. The imposed changes are in yellow colour in the manuscript. We do appreciate the wise comment and try to satisfy each comment accordingly.

RESPONSE TO THE COMMENTS

Question 1. Line 11 - rewrite sentence (This evaluation is determined a newly developed analytical probabilistic method for entire strain–life models of automobile suspension systems.)

Answer: The proposed technique is determined using a probabilistic method of the Gumbel distribution for strain–life models of automobile suspension systems. (Page 1, lines 12 and 13)

Question 2. Line 317 - Figure 4 caption does not mention what is (a) (b) and (c).

Answer: The caption of this figure is changed. This figure illustrates Strain signals measured for (a) rural, (b) campus, (c) highway road loads. (Page 10, Figure 6). (Please, see the revised manuscript).

Question 3. Why n=9? What constitutes a sample?

Answer: The probabilistic model was applied to 9 lives of the strain fatigue life models (each of rural, campus and highway road loads was allocated 3 lives based on the Coffin–Manson, Morrow and SWT models, separately), as shown in Table 3. These expressions are highlighted to revise the manuscript. (Page 13, line 415 to 417).

Question 4. Tables 3 and 4 - remove unnecessary decimal places.

Answer: Unnecessary decimal is removed from Table 3 and 4, as follows: (Pages 11 and 13). (Please, see the revised manuscript or .docx file (response to comments)).

Reviewer 3 Report

The paper presents a modification to the Coffin-Manson, Morrow and SWT strain life curves which takes into account the Gumbel probability density function. The approach is based on an algorithm of data fitting and it is overall an optimization algorithm. This approach may be useful in the case of low cycle fatigue lifetime estimation, but if we are dealing with mean strain or mean stress effect,  which was neglected by the authors beside the Morrow model is the reason why the strain life curves were way above experimental points. The non-Gaussian effect is not explained well enough, especially in terms of the influence on the fatigue life. The authors should focus on adding information on the effects of non-stationarity and non-gaussianity with more references. The material presented in the paper is not ground breaking, but is a good case of decent data analysis performed by the authors. In the reviewers opinion the paper should be published after minor changes discussed in this review as well as minor language changes.

Author Response

Dear Reviewer,

Thank you for your useful comments on the structure of our manuscript. We have modified the manuscript accordingly, and the corrections are listed as follows. The imposed changes are in yellow color in the manuscript. We do appreciate the wise comment and try to satisfy each comment accordingly.

RESPONSE TO THE COMMENTS

Question 1. This approach may be useful in the case of low cycle fatigue lifetime estimation, but if we are dealing with mean strain or mean stress effect, which was neglected by the authors beside the Morrow model is the reason why the strain life curves were way above experimental points.

Answer: The Morrow model is applied in order to calculate mean stress effect where components are subjected to compressive and tensile loadings [42].  . Mean stress is significantly affected when the elastic strain amplitudes are dominant [41]. Thus, the Morrow model predicted fatigue lives based on experimental data beyond the strain life curve where vehicle coil spring is subjected to tensile mean stress. In the case of the SWT model, the product of  and  parameters are remained constant for various combination of maximum stress and strain amplitude Thus, the SWT model predicts good assessment, where mean stress effect is applied under tensile mean stress cases in low cycle fatigue approach [45].  These explanations are added and highlighted in the revised version of the manuscript (page 14, line 444 to 453).

Question 2. The non-Gaussian effect is not explained well enough, especially in terms of the influence on the fatigue life. The authors should focus on adding information on the effects of non-stationarity and non-Gaussianity with more references.

Answer: Nieslony et al. [15] assessed fatigue life for non-Gaussian loading signals using a combination of the spectral method and the Dirlik method. In this purpose, the Dirlik and spectral methods were used to evaluate the experimental results obtained with the rainflow cycle technique. Results showed that non-Gaussian loadings can be assessed using the spectral method in the absence of mean stress effect. Cianetti et al. [16] represented a new correction coefficient to predict fatigue life for a mechanical component based on non-Gaussian stress in the frequency domain. The proposed procedure illustrated that the correction coefficient can be used to estimate fatigue damage on the basis of stress time histories affected when the kurtosis value is low. While, fatigue damage results were overestimated in the case of non-Gaussian stress because a new correction coefficient was required for implementation. However, relationship of non-stationarities and non-Gaussianities have been studied by Cesnik and Capponi in vibration fatigue analysis. The Gaussianity and stationarity are one of the important assumptions of the fatigue damage theory in frequency domain approach [17,18]. Capponi indicated that different rates of amplitude-modulated non-stationary excitation have a shorter fatigue life than the stationary excitation level for the dynamic structure’s response and dynamic loading.  These explanations are added and highlighted in the revised version of the manuscript (page 2, line 58 to 72).

Round 2

Reviewer 1 Report

REVIEW

on the article

Reza Manouchehrynia, Shahrum Abdullah, and Salvinder Singh Karam Singh

 Fatigue Reliability Assessment of an Automobile Coil Spring under Random Strain Loads using Probabilistic Technique

Summary. The article is devoted to the urgent issue of developing a mathematical model for predicting the reliability of coil springs under conditions of fatigue loading. The article has been substantially revised, many mistakes have been fixed, the meaning of the article looks better.

            However…

Specific comments.

 In equation 1 x is just the argument of a function, independent variable. It is not a random sample of a model.  Figure 4. It should look like this. So, the maximum of the function is in the point x=μ (see file) Equation 2. You should remove  and write, that by the change of variable you obtain new probability function (see file)

Because if you have then you have only part of function x≥μ

If you remove, then you have a dimensionless form of Gumbel distribution, and it’s OK.

Figure 5, line 265. The vertical axis is not the probability of failure, it is the probability density of failure. Figure 5 is also incorrect. Gumbel distribution from - to +. The probability density function has a maximum at point x=µ (see file) In concluding my review, I would advise the authors for the future (not in this article) to abandon the distribution of Gumbel. This distribution does not correspond to the physical meaning of the durability of the springs. The range from minus infinity to plus infinity shows the likelihood that durability can be negative, but this cannot be in principle.

Author Response

Dear Reviewer,

Thank you for your useful comments on the structure of our manuscript. We have modified the manuscript accordingly, and the corrections are listed as follows. The imposed changes are in yellow colour in the manuscript. We do appreciate the wise comment and try to satisfy each comment accordingly.

RESPONSE TO THE COMMENTS

SPECIFIC COMMENTS

Question 1. In equation 1 x is just the argument of a function, independent variable. It is not a randomly sample of a model.

Answer: In equation 1, x is changed to independent variable, (Page 5, line 161).

Question 2. Figure 4. It should look like this. So, the maximum of function is in the point .

Answer: Figure 4 is changed. Please see the revised manuscript or answer to comments in .docx file, (Page 5, line 164 to 166).

Question 3. Equation 2. 

Answer: Equation 2 is changed as below, that by the change of variable, the new probability function, Please see the revised manuscript or answer to comments in .docx file, (Page 6, line 176 to 179).

Question 4. Figure 5, line 265. The vertical axis is not the probability of failure; it is the probability density of failure.

Answer: The vertical axis in Figure 5 is changed to probability density of failure, (Page 8, line 265).

Question 5. Figure 5 is also incorrect. Gumbel distribution from -∞ to +∞. The probability density function has a maximum at point x=μ.

Answer: Figure 5 is changed. Please see the revised manuscript or answer to comments in .docx file, (page 8, line 266 to 268).

Question 6. In concluding my review, I would advise the authors for the future (not in this article) to abandon the distribution of Gumbel. This distribution does not correspond to the physical meaning of the durability of the springs. The range from minus infinity to plus infinity shows the likelihood that durability can be negative, but this cannot be in principle.

Answer: The suggestion will be considered for the future with limits to show the likelihood for durability.
